# Linking Self-Control to Negative Risk-Taking Behavior among Chinese Late Adolescents: A Moderated Mediation Model

**DOI:** 10.3390/ijerph19137646

**Published:** 2022-06-22

**Authors:** Zi-Qin Liang, Kai Dou, Jian-Bin Li, Yu-Jie Wang, Yan-Gang Nie

**Affiliations:** 1Department of Psychology and Research Center of Adolescent Psychology and Behavior, School of Education, Guangzhou University, Guangzhou 510006, China; ziqin.liang@studenti.unipg.it; 2Department of Philosophy, Social Sciences and Education, University of Perugia, 06123 Perugia, Italy; 3Department of Early Childhood Education, The Education University of Hong Kong, Hong Kong, China; lijianbin@eduhk.hk; 4Mental Health Education and Counseling Center, Guangdong Industry Polytechnic, Guangzhou 510300, China; psywyj@foxmail.com

**Keywords:** self-control, negative risk-taking behavior, regulatory focus, sense of power, adolescents

## Abstract

Negative risk-taking behaviors refer to voluntary behaviors that lead to more harm than good. Low self-control is a crucial predictor of adolescents’ negative risk-taking behavior, but its internal mechanisms require further exploration. To reveal the working process underlying the association between self-control and adolescents’ negative risk-taking behaviors, we investigated the mediation of regulatory focus and the moderation of sense of power. A total of 2018 students (37.6% males) from two universities in Guangzhou, China, participated in a survey that investigated their self-control, negative risk-taking behavior, regulatory focus and sense of power. The results revealed that after controlling for the adolescents’ sex and their parents’ educational level, prevention focus partially mediated the association between self-control and negative risk-taking behavior. Moreover, sense of power moderated the association between self-control and prevention focus. Furthermore, the association between self-control and negative risk-taking behavior through prevention focus was stronger among adolescents with a high sense of power than among those with a low sense of power. Therefore, our findings suggest that regulatory focus and sense of power might be the mechanisms that explain how self-control is related to negative risk-taking behavior. These results thus provide a foundation for the prevention of and intervention in adolescents’ negative risk-taking behavior.

## 1. Introduction

Risk-taking “involves voluntary choices for behaviors where outcomes remain uncertain” [1] (p. 353). Based on this, negative risk-taking behavior refers to the behavior that an individual engages in with the expectation of rewarding results while acknowledging the potential risks and their associated negative consequences [2]. In many cases, negative risk-taking behaviors impose more undesirable impacts on adolescent development than positive risk-taking behaviors (e.g., participating in challenging courses and sports), e.g., injuries, threats to physical and mental health, and even death [1]. Due to the unbalanced development of their socioemotional system and cognitive control system, adolescents lack sufficient self-control [3], and negative risk-taking behaviors reach a peak during adolescence [4].

According to problem behavior theory [5], an individual’s problem behavior (e.g., negative risk-taking behavior) is jointly determined by protective and risk factors, with protective factors reducing and risk factors increasing an individual’s propensity for engaging in problem behavior. Among the factors, self-control has been identified as a crucial predictor of risk-taking behavior in adolescents, and previous studies have found that low self-control increases negative risk-taking behavior [6,7,8]. However, less is known about the underlying mechanisms, such as the mediation process (i.e., how self-control is linked to negative risk-taking behavior) and moderation process (i.e., when self-control is linked to negative risk-taking behavior). Based on the promotion and prevention processes of self-control [9,10], we examined the idea that adolescents with good self-control will engage in less negative risk-taking behavior by mobilizing regulatory focus (promotion focus and prevention focus). In addition, we investigated sense of power—a disposition construct that affects the activation of the individual’s basic behavioral system and regulates an individual’s goal-directed motivation—as a moderator of the association between self-control and regulatory focus. These proposed variables belong to the personality-related structures under the conceptual framework of protection in problem behavior theory, entailing they would decrease the likelihood of engaging in problem behavior. This study contributes to the salient literature in two ways. Theoretically, the present study extends the problem behavior theory by refining *how* personality constructs are related to an individual’s problem behavior. Practically, the findings highlight leverage points that can be targeted to reduce negative risk-taking behavior in adolescents.

### 1.1. Self-Control and Negative Risk-Taking Behavior

Self-control refers to an individual’s ability to alter his or her dominant cognitive, emotional, and behavioral reactions to follow social norms and to pursue long-term goals [11]. According to problem behavior theory, protective factors account for a diminished likelihood of the occurrence of problem behaviors by directly mitigating or buffering the impact of exposure to risk factors [5,12]. Good self-control is a protective factor in the personality system, which can promote good internal inhibitions and help adolescents establish their own behavioral norms to prevent and reduce the occurrence of negative risk-taking behavior. The dual-system model of adolescent risk taking can also be used to explain the association between self-control and adolescents’ risk-taking behavior. According to the dual-system model, adolescents are particularly vulnerable to risk-taking via the confluence of the developmental patterns of their socioemotional and cognitive control systems, that is, their relatively high responsiveness to reward combined with their relatively weak self-regulation [3]. Adolescents’ risk-taking is driven by the enhanced activation of sensation seeking and reward sensitivity during adolescence [4]. As a form of cognitive control, a high level of self-control can help adolescents use better inhibitory control to plan and monitor to prevent or avoid engaging in negative risk-taking behavior [3]. Conversely, adolescents with low self-control have more negative risk-taking behavior [13,14]. A general theory of crime, which was proposed by Gottfredson and Hirschi [13], suggests that lack of self-control is one of the key factors leading to juvenile delinquency. For instance, a study on how children’s self-control predicts lifetime smoking found that low childhood self-control makes children more susceptible to tobacco use during adolescence and to becoming smokers, even leading to an elevated risk of smoking for many decades [6]. Holmes et al. [7] found that higher self-control trajectories from mid-childhood to late childhood are associated with lower risk-taking in adolescence. Moreover, some intervention studies found that the enhancement of control ability can help to reduce the risk that adolescents will participate in negative risk-taking behaviors, such as smoking, alcohol and drug use, unhealthy eating, or forming friendships with deviant peers [8]. Overall, these studies suggest that as problem behavior theory and the dual-system model predict, higher self-control helps reduce or prevent adolescents’ negative risk-taking behavior. However, these theoretical accounts provide little explanation for how self-control would be related to negative risk-taking behavior.

### 1.2. Regulatory Focus as a Mediator

The promotive and preventive mechanisms of self-control offer a framework to interpret how self-control is related to negative risk-taking behavior [15,16]. The promotive mechanism refers to the process of mobilizing positive thoughts, emotions, and behaviors, while the preventive mechanism refers to the process of demobilizing negative thoughts, emotions, and behaviors. Both processes share similar functions—to facilitate the actualization of goals and the adherence to rules [10,16]. Based on this framework, we examined regulatory focus as a mediator in the present study.

Regulatory focus encompasses the following two separate motivational orientations to guide adolescents’ goal-pursuit behaviors: promotion focus and prevention focus [17]. These two orientations are derived from individuals’ subjective history of success in promoting and preventing the achievement of goals in the past, which will energize and direct them to adopt different strategies to approach new task goals [18]. Promotion focus refers to the utilization of approaching strategies to attain goals, while prevention focus refers to the use of vigilant and avoidant strategies to attain goals [17,18]. Based on this, Higgins et al. developed the Regulatory Focus Questionnaire (RFQ) to measure individuals’ subjective history of promotion success and prevention success, respectively, and regulatory focus also influences the critical process of selecting means to attain task goals [18,19]. According to the potential positive or negative outcomes, self-control will enhance different motivational orientations to adopt corresponding behavioral strategies [9,20]. Specifically, for individuals with high self-control, their promotion focus strategies will be more likely to be activated when they encounter rewarding or positive results. They will be more concerned with gaining positive benefits and long-term goals, and their promotion focus may serve as a motivation to continuously seek new ways to achieve goals [9]. Moreover, individuals with high self-control will also activate their prevention focus strategies when they encounter negative results that do not meet their desirable goals. So, they will be more sensitive and vulnerable to the negative information that a behavioral result may produce, thus showing stronger protective and vigilant motivation to avoid mismatched behavior or target deviation [20]. Thus, people with high self-control are likely to utilize both promotion and prevention focus strategies to achieve goals, but the focus activated depends on whether the outcomes/goals are desirable or not. Promotion focus will be activated when the outcomes/goals are desirable, whereas prevention focus will be activated when the outcomes/goals are undesirable.

As mentioned above, negative risk-taking behaviors may have uncertain outcomes [1], which suggests that adolescents who are promotion- or prevention-focused will show different behavioral tendencies when facing risky stimuli. Adolescents with promotion-focus strategies tend to pay more attention to the needs of improvement, growth, and nourishment, which makes adolescents take risks to achieve positive outcomes [21]. Adolescents with prevention focus strategies are more concerned with safety, fulfilling their duties and security needs. They will not actively seek risks, even in a state of no loss or possible benefit [22]. Empirical studies have shown that adolescents with a promotion-focus strategy have a high level of risk decision-making willingness [21,22]. Because they are more willing to change and take chances to fulfill their needs in undesirable situations, when a situation is clearly good, they will maintain the status quo by being vigilant. Similarly, adolescents with a prevention-focus strategy show a conservative orientation [23]. Hence, when adolescents face negative risk-taking behavior, their prevention focus strategies play a great role and may help them prevent themselves from generating undesirable outcomes. Moreover, their promotion-focus strategies may also play a role because of the potential risks with the expectation of rewarding results. Taken together, these rationales and the evidence suggest that promotion focus and prevention focus would be likely to mediate the association between self-control and adolescents’ negative risk-taking behavior, respectively.

### 1.3. Sense of Power as a Moderator

Sense of power refers to the perceived relative ability of an individual to influence other people in a social relationship [24]. Individuals with a high sense of power believe that they have power in interpersonal relationships and have a greater ability to influence others [24]. They are more likely to act on internal driving forces (e.g., their own values and attitudes) and are less affected by external cues (especially peers) [25]. High-power individuals tend to show better self-control than low-power individuals [26]. Because they have more advantages when balancing goals and adjusting their attention [27], they control their impulsiveness or adopt delayed gratification strategies to achieve long-term goals. Heller et al. [26] showed that when a task requires the participation of self-control, a high sense of power can increase self-control to sufficiently initiate and maintain behavior, thereby promoting the completion of the task.

Adolescents with a high sense of power are more likely to achieve goals by regulating their motivation to influence their behaviors, that is, by enhancing their motivation to approach what they want or/and to avoid what they do not want [28,29]. On the one hand, amid positive outcomes, they will be sensitive to gains and rewards and tend to enhance their approach motivation. On the other hand, amid negative outcomes, they will be cautious, preventing new and correcting existing negative states to enhance their avoidance motivation and promote the realization of their own goals, i.e., they will tend to avoid negative behaviors. Accordingly, high-power adolescents can regulate two types of behavioral motivations, approaching and avoiding, which are similar to the two motivational strategies of promotion- and prevention-focus strategies. In other words, high levels of power can enhance adolescents’ motivation to promote or/and prevent. Yang et al. [28] found that, in the face of positive outcomes, high power individuals control the motivation to approach or avoid due to personal ideals and desires and that they are more inclined to enhance promotion-focus strategies; however, low-power individuals control their motivation to approach or avoid due to responsibilities and obligations, and they are more inclined to enhance prevention-focus strategies. In addition, Hoogervorst et al. [30] showed that power is often associated with more social responsibilities in collectivist cultures; that is, high-power individuals have a high sense of responsibility [31] and are more likely to increase prevention motivation to protect the interests of most people. Scheepers et al. [32] found that to maintain and protect their existing situation, high-power holders tend to make prevention-focused decisions. Deng et al. [31] point out that, in insecure situations, such as the salience of death and threats to self-worth, individuals with high-power would adopt defensive strategies, such as avoidance motivation. Thus, when the outcomes are negative, adolescents with a high sense of power are likely to increase prevention focus and use preventive strategies to maintain a positive status quo and avoid undesirable factors. Taken together, these rationales and the evidence suggest that sense of power is likely to moderate the association between self-control and regulatory focus. Moreover, a high sense of power further compels adolescents with high self-control to increase their motivation for prevention focus or/and promotion focus, thereby reducing their negative risk-taking behavior.

### 1.4. The Current Study

In this study, we sought to understand the underlying mechanisms of how self-control is related to negative risk-taking behavior in adolescents. To address this issue, we proposed regulatory focus as a mediator and sense of power as a moderator. Therefore, we examined a moderated mediation model, as shown in Figure 1. We hypothesized that (1) self-control would be negatively related to adolescents’ negative risk-taking behavior, (2) regulatory focus would mediate the association between self-control and adolescents’ negative risk-taking behavior, that is, promotion and prevention focus would mediate this association, respectively, (3) sense of power would moderate the association between self-control and regulatory focus (i.e., promotion and prevention focus), with a high sense of power strengthening the association between self-control and regulatory focus, and (4) the indirect associations between self-control and negative risk-taking behavior via regulatory focus (i.e., promotion and prevention focus) would vary as a function of sense of power, with the mediation effect of regulatory focus being more pronounced for adolescents with high-power than for those with low-power. In addition, some demographic variables (e.g., adolescents’ sex, parents’ educational levels) were included as covariates because previous studies found that males are more prone to engaging in risk-taking behavior than females during adolescence [33,34] and that parents’ education levels are negatively correlated with risk-taking behavior [35].

## 2. Materials and Methods

### 2.1. Participants and Procedure

Data were collected from two universities in Guangzhou, China. A total of 2018 undergraduates (759 males, 1259 females) joined the study and completed the survey with paper and pencil. Among them, 508 students (25.2%) were freshmen, 396 sophomores (19.6%), 554 juniors (27.5%) and 560 seniors (27.8%). Regarding participants’ parents’ level of education, 672 fathers (33.3%) and 855 mothers (42.4%) had received at least a primary school degree, 959 fathers (47.5%) and 851 mothers (42.2%) had finished secondary school, 362 fathers (17.9%) and 302 mothers (15.0%) had earned a bachelor’s degree, and 25 fathers (1.2%) and 10 mothers (0.5%) had earned a master’s degree or above.

Due to the cross-sectional nature of the design, we implemented several procedures to mitigate the common method bias and social desirability bias [36]. Specifically, all participants were assured that their responses would be confidential and that the data they provided would be used only for research purposes. Additionally, the order of the questionnaires and items was randomized. All university students provided written consent before participating in the study. They completed the questionnaires under the guidance of administrators who were trained graduate students majoring in psychology. The questionnaire was collected on site, and no compensation was given. All procedures involving human participants were reviewed and approved by the research ethics committee in the school of education at the corresponding author’s university.

### 2.2. Measures

#### 2.2.1. Self-Control

Self-control was measured with the Brief Self-Control Scale (BSCS) developed by Tangney et al. [37]. This scale includes 13 items rated on a 5-point scale from 1 (*not like me at all*) to 5 (*very much like me*), with a higher mean score (with 9 items reverse-coded) indicating better self-control. The BSCS presented good psychometric properties across different Chinese samples [38]. Sample items include “I do certain things that are bad for me if they are fun” (reverse-scored) and “I am good at resisting temptation” (see Appendix A). The Cronbach’s alpha of this scale was 0.80 in this study.

#### 2.2.2. Regulatory Focus

Regulatory focus was measured with the Regulatory Focus Questionnaire (RFQ) developed by Higgins et al. [18]. This scale was validated in the Chinese context, showing good psychometric properties [39]. Our scale consisted of two subscales that measure promotion focus and prevention focus. It included 11 items rated on a 5-point scale from 1 (*never or seldom*) to 5 (*very often*), with a higher mean score of each dimension indicating higher levels of promotion/prevention focus. Sample items include “Compared to most people, are you typically unable to get what you want out of life?” and “Did you get on your parents’ nerves often when you were growing up?” (see Appendix A). In this study, the Cronbach’s alpha of promotion focus and prevention focus were 0.79 and 0.80, respectively.

#### 2.2.3. Sense of Power

Sense of power was measured with the Personal Sense of Power Scale (PSPS) developed by Anderson et al. [24]. This scale was validated in the Chinese context, showing good psychometric properties [28]. Our scale included 8 items rated on a 7-point scale from 1 (*strongly disagree*) to 7 (*strongly agree*), with a higher score (with 4 items reversely coded) indicating a greater sense of power. Sample items include “Even when I try, I am not able to get my way” (reverse-scored) and “I can let my friends listen to me” (see Appendix A). The Cronbach’s alpha of this scale was 0.73 in this study.

#### 2.2.4. Negative Risk-Taking Behavior

Three subscales of the Adolescent Risk-taking Questionnaire-Risk Behavior Scale (ARQ-RB; [40]), namely, rebellious behavior (4 items, e.g., smoking, getting drunk), antisocial behavior (3 items, e.g., cheating, overeating), and reckless behavior (4 items, e.g., leaving school, having unprotected sex), were used to measure adolescents’ negative risk-taking behavior (see Appendix A). Our scale included 11 items rated on a 5-point scale, from 0 (*never*) to 4 (*very often*). A mean score of the three subscales was calculated, and a higher score indicated more engagement in negative risk-taking behavior. The ARQ-AB was validated in the Chinese context, showing good psychometric properties [41]. The Cronbach’s alpha of this scale was 0.87 in this study.

#### 2.2.5. Covariates

Adolescents’ sex (0 = *female*, 1 = *male*) and parents’ educational levels (used as a proxy for socioeconomic status (SES); 0 = *primary school degree or below*, 1 = *secondary school degree*, 2 = *bachelor’s degree*, 3 = *graduate degree or above*) were included as covariates since previous studies found significant associations between these demographic variables and negative risk-taking behavior [33,34,35].

### 2.3. Data Analysis

SPSS 22.0 and Mplus 8.3 were used to test the primary hypotheses. Descriptive statistics and bivariate correlations were conducted using SPSS 22.0 (IBM, Armonk, NK, USA) to test the levels of and associations among the key variables. Then, the mediation model and moderated mediation model were tested separately via path analysis using Mplus 8.3 (Muthén & Muthén, Los Angeles, CA, USA). Adopting a three-step procedure, we first examined the effect of self-control on risk-taking behavior (Hypothesis 1). Then, we examined the mediation model to test Hypothesis 2. Next, we integrated sense of power to test for moderation and further moderated mediation [42]. Specifically, in the mediation model, self-control was the independent variable, promotion and prevention focus were two parallel mediators, and negative risk-taking behaviors were the outcomes. We examined the mediation model in two steps. In the first step (i.e., the total effect model), we investigated the association between self-control and negative risk-taking behaviors without including the mediator. In the second step (i.e., the indirect effect model), we examined the association between self-control and negative risk-taking behaviors by including the mediator. The findings thus inform how the changes in the magnitude of the “self-control–negative risk-taking behaviors” link are a function of the mediators. We also compared the differences between the two mediation pathways to distinguish the relative importance of promotion focus and prevention focus. Based on the mediation model, sense of power was added as the moderator. In the moderated mediation model, self-control and sense of power were mean-centered. When a significant moderation effect was found, we continued via a simple slope test by high (+1 *SD*) and low (−1 *SD*) levels of sense of power. We also examined whether the mediation effect of regulatory focus was significant by using different levels of sense of power.

Given that the bootstrapping technique has several advantages (e.g., higher statistical power) over the traditional approaches for examining mediation models [43], we used bootstrapping (*N* = 5000) and 95% confidence intervals (95% CI) to judge the significance of the indirect effect [42]. As long as the 95% CI excludes 0, a significant mediation effect will be tenable. Thus, the following indices were used to evaluate overall model fit [44]: the comparative fit index (CFI, no less than 0.90), the Tucker–Lewis index (TLI, no less than 0.90), the root-mean-square error of approximation (RMSEA, no larger than 0.08) with its 90% CI, and the standardized root-mean-square residual (SRMR, no larger than 0.08).

## 3. Results

### 3.1. Descriptive Statistics

Means, standard deviations, and bivariate correlations are shown in Table 1. Among the variables, self-control was positively related to promotion focus (*r* = 0.42, *p* < 0.001), prevention focus (*r* = 0.27, *p* < 0.001), and sense of power (*r* = 0.36, *p* < 0.001) but negatively related to negative risk-taking behavior (*r* = −0.22, *p* < 0.001). Promotion focus was positively related to sense of power (*r* = 0.44, *p* < 0.001) but negatively related to negative risk-taking behavior (*r* = −0.11, *p* < 0.001). Prevention focus was positively related to sense of power (*r* = 0.16, *p* < 0.001) but negatively related to negative risk-taking behavior (*r* = −0.16, *p* < 0.001). These findings provided preliminary support for our hypothesized associations.

### 3.2. Testing for Mediation Effects

We first examined the total effect model of the association between self-control and negative risk-taking behavior. Consistent with Hypothesis 1, self-control was negatively associated with negative risk-taking behavior (*β* = −0.18, *p* < 0.001) after controlling for adolescents’ sex, father’s education and mother’s education. Next, we examined the indirect effect model by including regulatory focus in the model, thereby controlling for the covariates (Figure 2). The model fit indices were good, χ2 = 8.94, *df* = 3, *p* < 0.05, RMSEA = 0.03 (90% CI = [0.009, 0.056]), CFI = 0.99, TLI = 0.97, SRMR = 0.01. The results showed that self-control was positively associated with promotion focus (*β* = 0.45, *p* < 0.001) and prevention focus (*β* = 0.32, *p* < 0.001). High levels of prevention focus were associated with less negative risk-taking behavior (*β* = −0.06, *p* < 0.01), but promotion focus was not significantly associated with negative risk-taking behavior (*β* = −0.01, *p* = 0.59).

Finally, the results of the bias-corrected bootstrapping test of the indirect effects indicated that the total mediation effect was −0.02 (95% CI = [−0.050, −0.004]), *p* < 0.05. As shown in Table 2, we found that the association between self-control and negative risk-taking behavior was mediated by prevention focus (*β* = −0.02, 95% CI = [−0.034, −0.009]) but not by promotion focus (*β* = −0.01, 95% CI = [−0.031, 0.019]). Furthermore, a comparison of the mediating effect suggested that there was no significant difference between promotion focus and prevention focus (*β* = 0.01, 95% CI = [−0.015, 0.039]).

### 3.3. Testing for Moderated Mediation Effect

We performed a linear regression analysis to examine whether sense of power moderated the association between self-control and regulatory focus and whether self-control and sense of power were mean-centered. As shown in Table 3, the results indicated that sense of power moderated the association between self-control and prevention focus (*β* = 0.09, *p* < 0.001), but it did not moderate the association between self-control and promotion focus (*β* = 0.02, *p* = 0.45). The results of the simple slope test (Figure 3) indicated that the association between self-control and prevention was stronger among undergraduates who reported a high sense of power (*β* = 0.43, *p* < 0.001) than among those who reported a low sense of power (*β* = 0.15, *p* = 0.10).

A moderated path approach was used to test the moderated mediation hypotheses [42]. The results showed that, when the sense of power was high (i.e., one *SD* above the mean), the indirect effect of prevention focus was significant (*β* = −0.02, *SE* = 0.01, 95% CI = [−0.036, −0.011]), but it was not significant (*β* = −0.01, *SE* = 0.01, 95% CI = [−0.021, 0.004]) when the sense of power was low (i.e., one *SD* below the mean). The difference between the indirect effects of low and high sense of power was 0.01, with a 95% base-corrected bootstrap confidence interval of 0.001 to 0.025. In sum, these results indicated that self-control was more strongly associated with negative risk-taking behavior through prevention focus when the sense of power was high.

## 4. Discussion

Adolescence is a developmental period of increased negative risk-taking behavior [4]. Engaging in negative risk-taking behavior is related to many negative outcomes that restrain adolescents’ positive development [1]. Thus, understanding the factors and the underlying mechanisms of adolescents’ negative risk-taking behavior is important. Using problem behavior theory [5] and the dual-system model [3], this study therefore tested a moderated mediation model to reveal the mechanism underlying the association between self-control and negative risk-taking behavior in adolescents. Our results showed that self-control was negatively related to negative risk-taking behavior in adolescents and that this association was mediated by prevention focus. In addition, sense of power moderated the association between self-control and prevention focus and the mediation effect of prevention focus. Specifically, the association between self-control and negative risk-taking behavior through prevention focus was stronger among adolescents with a high sense of power than among those with a low sense of power.

### 4.1. Self-Control and Negative Risk-Taking Behaviors

This study demonstrates that adolescents with high self-control have less negative risk-taking behavior, which is consistent with previous studies [6,7,8]. It also provides evidence for the prediction results of problem behavior theory and the dual-systems model [3,4,5,12]. Hence, the main cause of negative risk-taking behavior in adolescence is a lack of impulse control ability. Adolescents with high self-control can effectively overcome any impulse or automatic reaction caused by temptation. They will control their own behavior and inhibit the occurrence of problem behavior. However, adolescents with low self-control pursue the immediate pleasure brought by temptation and ignore any potential negative consequences, leading to the generation of negative risk-taking behavior.

### 4.2. The Mediating Role of Regulatory Focus

More importantly, we find that prevention focus, as a behavioral motivation to avoid adverse outcomes, plays a significant mediating role in the “self-control–reduced negative risk-taking behaviors” link. Accordingly, adolescents with high self-control tend to adopt prevention-focus strategies (rather than promotion focus strategies) to avoid or prevent negative risky behavior. This result is consistent with previous research findings [45,46]. It further exhibits the “promotive–preventive” framework of self-control, where the prevention mechanism of self-control plays a role in reducing the possibility of adolescents’ negative risk-taking behavior. The mediating role of promotion focus was not established, and the promotive mechanism of self-control does not influence negative risk-taking behavior. This is because the promotion mechanism of self-control mobilizes positive thoughts, emotions and behaviors in the process of achieving goals [10], and promotion focus is oriented toward obtaining positive outcomes as its ultimate goal [17,21]. Both facilitate positive results. However, even if negative risk-taking behaviors bring immediate stimulation and pleasure, they have irreversible consequences that damage adolescents’ health [1]. The harm caused to adolescents is far greater than any benefits and is not conducive to their pursuit of positive goals. Adolescents with high self-control cannot initiate the motivation of promotion focus to participate in negative risk-taking behaviors. Therefore, only prevention focus has a mediating effect between self-control and negative risk-taking behavior.

### 4.3. The Moderating Role of Sense of Power

Another important finding is that the sense of power significantly moderates the effect of self-control on prevention focus; that is, the mediation effect from self-control to negative risk-taking behavior through prevention focus is stronger among adolescents with a high sense of power. For example, high-power adolescents approach their ultimate goal by increasing their preventive motivation to avoid anything that they do not want [28]. While previous studies mostly focused on the role of power in promoting a positive behavior framework [27], this study enriched the understanding of the important role of power in reducing a negative behavior framework and has shown that the sense of power is a protective factor in problem behavior theory. In this study, the sense of power did not moderate the association between self-control and promotion focus, which may be related to a negative behavioral outcome. Previous studies found that to obtain more resources, adolescents with a high sense of power enhance the role of self-control in increasing promotion focus; thus, individuals can be motivated by discovering opportunities and seeking rewards [28,31]. However, when the outcomes/goals are negative risk-taking behaviors, their potential threats are far greater than their positive benefits, which not only consume many resources but also, amid improper control, endanger their physical and mental health [1]. As mentioned above, self-control resists the temptation of negative risk-taking by initiating a preventive focus [45,46]. Hence, adolescents with a high sense of power will enhance the role of self-control in increasing prevention focus, thereby maintaining their current positive state and preventing the emergence of new negative states, which helps reduce the probability of negative risk-taking behavior. Therefore, adolescents with a high sense of power enhance the mediation effect between self-control and negative behavior through prevention focus.

### 4.4. Implications

This study has both theoretical and practical implications. Previously developed theories (problem behavior theory and dual-system model) suggest that self-control is only a predictor; they do not explain how self-control relates to negative risk-taking behavior. This study fills this knowledge gap by revealing that high self-control reduces negative risk-taking behavior through prevention focus. Moreover, this mechanism has a more significant effect on adolescents with a high sense of power than on those with a low sense of power. In practical terms, the current findings identify several leverage points to reduce adolescents’ negative risk-taking behavior. First, schools and families can train and cultivate adolescents’ self-control ability, for example, via mindfulness training [47]. Second, parents and educators should coach and guide adolescents to set goals and encourage them to use prevention strategies to maintain existing consequences and ongoing efforts [48]. Moreover, parents can encourage adolescents to make their own decisions, and schools can provide more courses and activities to enhance adolescents’ sense of power by, for example, training students in leadership skills and key social and emotional learning competencies [49].

### 4.5. Limitations and Future Directions

Notably, this study has some limitations. First, the cross-sectional design made it impossible to deduce the directionality and causality among variables. To address this issue, we tested a number of competing models and found that our chosen model showed the best fit. Nevertheless, future studies may need to employ more sophisticated research designs (e.g., longitudinal and experimental designs) to further examine this topic. Second, although the indirect effects were statistically significant in the current study, the effect sizes were small. The sample size and the existence of potential confounding variables are the possible reasons for these results. More studies on this subject are desirable in the future. For example, future studies could consider the impact of risk perception, which has been identified as a significant personal factor influencing risk-taking behavior [50,51]. Third, this study focused only on negative risk-taking behavior; we did not include positive risk-taking behavior (e.g., participating in challenging courses and sports). Given that both negative and positive risk-taking behavior involve risk assessment and enactment [52], future studies may examine whether both types of risk-taking share similar predictors and working mechanisms. In addition, it is necessary to develop more culture-specific questionnaires to assess risk-taking behavior. Fourth, only a self-report questionnaire was used, which might inflate the associations among the study variables. To reduce such bias, we adopted some procedure controls when implementing the study (e.g., emphasis on anonymity and confidentiality). Nevertheless, it is strongly suggested that future research should employ multiple-report informants (e.g., peer reports and parent reports) to triangulate each variable to assess the constructs more objectively. Despite these limitations, the current findings contribute to the literature on adolescent risk-taking and its developmental mechanisms.

## 5. Conclusions

In conclusion, this study found that high levels of self-control predict a decrease in adolescents’ negative risk-taking behavior by increasing their motivation for prevention focus. Furthermore, this association would be stronger among adolescents with a high sense of power. These findings provide a theoretical foundation for the prevention and intervention of adolescents’ negative risk-taking behavior. We suggest that attention should be focused on the adolescents’ negative risk-taking behavior, particularly in late adolescence. The probability of negative risk-taking behavior could be reduced by increasing one’s level of self-control, sense of power, and by developing positive goals or encouraging the use of prevention focus strategies when faced with negative consequences.

## Figures and Tables

**Figure 1 ijerph-19-07646-f001:**
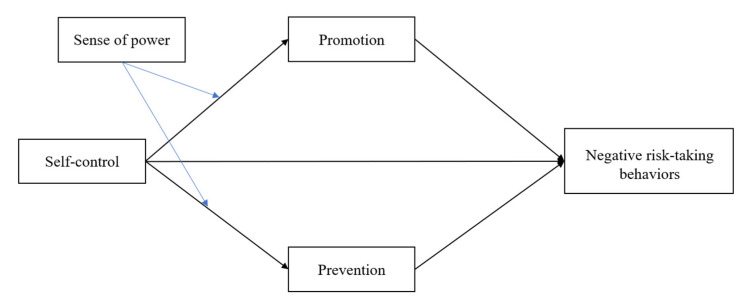
The conceptual model.

**Figure 2 ijerph-19-07646-f002:**
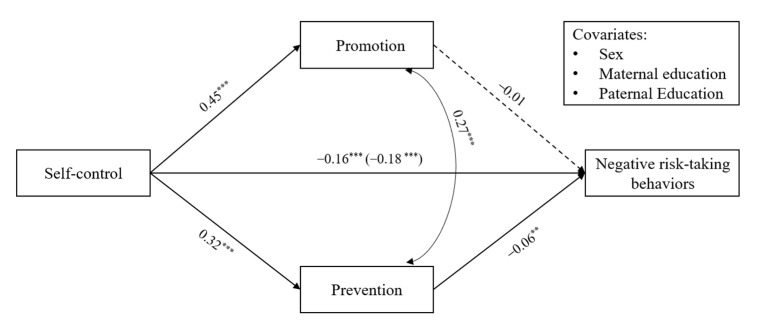
The mediating effect of regulatory focus in the relation between self-control and negative risk-taking behaviors. Note: Standardized estimates are presented; Values in parentheses refer to the effect without the mediator; ** *p* < 0.01, *** *p* < 0.001; Dashed line indicates a non-significant coefficient.

**Figure 3 ijerph-19-07646-f003:**
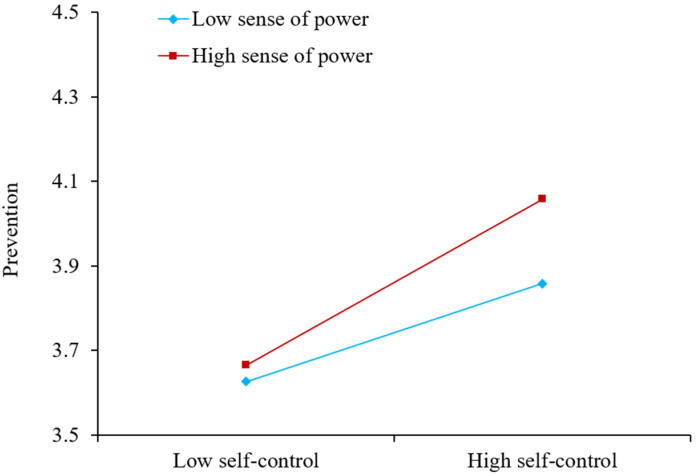
The moderating effect of sense of power in the relation between self-control and prevention focus.

**Table 1 ijerph-19-07646-t001:** The means, standard deviations, and bivariate correlations of the study variables.

	*M*	*SD*	1	2	3	4	5	6	7	8
**Control variables**										
1 Sex (male)	37.6%	—	—							
2 Father’s Education	2.3	1.9	0.00	—						
3 Mother’s Education	2.0	1.7	−0.00	0.75 ***	—					
**Independent variable**										
4 Self-control	3.2	0.6	−0.06 **	0.00	0.02					
**Mediating variables**										
5 Promotion	3.4	0.6	0.01	0.08 ***	0.08 ***	0.42 ***				
6 Prevention	3.7	0.7	−0.15 **	−0.06 **	−0.04	0.27 ***	0.33 ***			
**Moderating variable**										
7 Sense of power	4.5	0.8	−0.02	0.03	0.05 *	0.36 ***	0.44 ***	0.16 ***		
**Dependent variables**										
8 Negative risk-taking behaviors	0.5	0.5	0.20 **	0.02	0.01	−0.22 ***	−0.11 ***	−0.16 ***	−0.06 *	

Note: * *p* < 0.05, ** *p* < 0.01, *** *p* < 0.001.

**Table 2 ijerph-19-07646-t002:** The specific indirect effect for each indirect pathway in the mediation model based on the bias-corrected bootstrapped estimates.

Specific Pathways Tested in the Model	Bias-Corrected Bootstrapped Estimates for the Effects			
	*β*	*SE*	95% CI	*B*
**Direct pathway**				
**Self-control→Negative risk-taking behaviors**	**−0.18**	**0.02**	**[−0.25, −0.16]**	**−0.16**
**Indirect pathway**				
Self-control→Promotion→Negative risk-taking behaviors (ind1)	−0.01	0.01	[−0.03, 0.02]	−0.01
**Self-control→Prevention→Negative risk-taking behaviors (ind2)**	**−0.02**	**0.01**	**[−0.03, −0.01]**	**−0.02**
Difference = ind1 − ind2	0.01	0.01	[−0.02, 0.04]	0.01

Note: Significant effects were bolded.

**Table 3 ijerph-19-07646-t003:** Regression results for the analysis of the moderated mediation effect of self-control on negative risk-taking behaviors.

	Promotion as Dependent Variable		Prevention as Dependent Variable	
	M1	M2	M3	M4
**Control variables**				
Sex	0.04	0.04	−0.20 ***	−0.20 ***
Father’s education	0.02	0.02	−0.02	−0.02
Mother’s education	0.01	0.01	0.00	0.00
**Independent variable**				
Self-control	0.45 ***	0.32 ***	0.32 ***	0.28 ***
**Moderating variable**				
Sense of power		0.26 ***		0.08 ***
**Interaction term**				
Self-control × Sense of power		0.02		0.09 ***
*R* ^2^	0.18	0.27	0.09	0.11
Δ*R*^2^		0.09		0.02
*F*	111.01 ***	126.32 ***	53.29 ***	40.27 ***

Note: *N* = 2018; Sex was coded as 0 (female) and 1 (male); Education was coded as 0 (Primary school degree or below), 1 (Secondary school degree), 2 (Bachelor’s degree) and 3 (Graduate degree or above); M1 = model 1; M2 = model 2; M3 = model 3; M4 = model 4; The *β* values are standardized regression coefficients; *** *p* < 0.001.

## Data Availability

The data presented in this study are available on request from the corresponding author.

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
