# Peer review of "Linking Self-Control to Negative Risk-Taking Behavior among Chinese Late Adolescents: A Moderated Mediation Model"

_ijerph, 2022, doi:10.3390/ijerph19137646_

Round 1

Reviewer 1 Report

This paper contributed greatly to the literature body analyzing self-control and risk-taking behavior. However, the authors may want to consider the following questions in the study.

My primary concern of the study is the association among all variables. Why does this study want to select regulatory focus and sense of power for investigating the relationship between self-control and risk-taking behavior? The empirical studies have shown a significant association between self-control and risk-taking behavior. Will these new variables serve as a fresh perspective to interpret risk-taking decisions? Or the model still relies on the levels of self-control to predict risky behavior. To my understanding, the levels of self-control are stable after a certain age, which means an individual who has low self-control will commit more delinquency, and a person who has high self-control will not in their lifetime. Can this study use these new mediators and moderators to explain a different scenario (for example, a high self-control person will be more likely to commit risky behavior when their promotive focus is high)?

The discussion section gave a more precise explanation of the connections among variables. I would suggest using similar phrases such as prevention focus strategy in the literature review section so that readers can understand the research subjects.

1.1

Self-Control and Negative Risk-Taking Behavior

Important literature was missing. Gottfredson and Hirschi's theory talks about how self-control influences delinquency. Although you already mentioned some theoretical frameworks, you should add this piece to your literature review.

Gottfredson, Michael, and Travis Hirschi. 1990. A general theory of crime. Stanford, CA: Stanford Univ. Press

Line 71: The dual-system model

What does this refer to? The personality system? Please clarify.

Line 72: According to this model…

It's not clear to me which model that you are referring to.

Line 94: reference 10

I can't tell that this reference is related to the relationship between self-control/mechanism of self-control and negative risk-taking behavior.

Line 97-98

Need references for the promotive mechanism and preventive mechanism.

Line 174-175

This sentence is the opposite of the previous statement (160-161).

I didn't see the connection between the sense of power and negative risk-taking behavior. All your references focus on achieving goals and positive outcomes. How does it imply the negative consequences?

I have only one comment on the methods and results sections. Did the study conduct a multicollinearity test to ensure no collinearity issue among the variables? You can show the VIF value in the text.

Author Response

Thank you for the hard work devoted to our work. We also thank you for the positive comments. We have responded the comments point-by-point. Please see the attachment.

Reviewer 2 Report

This study investigated the mediation of regulatory focus and the moderation of sense of power to reveal the working process underlying the association between self-control and adolescents' negative risk-taking behaviors. The topic is interesting and the manuscript was well written. Some comments for the authors to improve the quality of the manuscript are provided as follows.

  1. In introduction part, risk-taking behavior has been widely investigated in different contexts. The authors are suggested to summarize the recent research on risk-taking behavior of people including workers, students, and the public in a table, which can show the novelty of this study.
  2. The hypotheses of the conceptual model should be clearly stated in the manuscript.
  3. Please add references to the moderation and mediation analysis procedures. Also, the procedures are not clear. Please revise them.
  4. I would like to see the content of measurement items in a table.
  5. Please provide standardized path coefficients rather than unstandardized ones in Table 2.
  6. What do M1, M2, M3 and M4 mean in Table 3?
  7. Risk perception may be an important factor that influences the negative risk-taking behavior of Chinese Late Adolescents as risk perception has been found to be significant in influencing risk-taking behavior of construction workers (Man et al, 2017, 2021). The authors may discuss this point in future research opportunities and limitations.

Author Response

(The authors gave the same response as above.)

Reviewer 3 Report

The present work describes a well-designed and well-developed study that mainly relates self-control with risk in university students' decision making, trying to identify significant mediators between the two. A very respectable sample size is used, to which boostrapping is also applied. The descriptive and inferential statistical analyses are correctly presented and well executed, presented and interpreted, making a complete discussion with pertinent conclusions that respond to the hypotheses initially proposed. The limitations of the study are logical and honest and future work is suggested.

Perhaps the fact of referring to the participant sample as adolescents when they are actually university students may generate some confusion about their true age range and maturity, so revising that concept may be pertinent. The section on conclusions is excessively brief as it has already been included in the discussion section, so it is worth considering devoting a single section to discussion and conclusions or, if kept separate, concentrating the conclusions in the section reserved for this purpose in a clearer way. The recommendations derived from the conclusions may be excessively simple and immediate considering the educational complexity inherent in stages where personality maturity is acquired, both in the school and family contexts. In any case, they are coherent with the conclusions reached, although they would probably require more in-depth strategies.

Congratulations, great job.

Author Response

(The authors gave the same response as above.)

Reviewer 4 Report

This manuscript describes theory-based associations between self-reported measures of risk-taking, self-control, regulatory focus and risk-taking behavior in a sample of young Chinese adults.  I was very impressed with the writing.  Nearly every section of the paper is well-argued and the organization overall was apparent. In addition, the analyses were appropriate and the conclusions were sensible and not overstated.  As such, I have only modest suggestions for improvement.

One exception to the clarity in the writing concerns discussion of regulatory focus.  Throughout both the introduction and discussion, the authors often conflate promotion and prevention focus.  That is, they discuss individuals as being either high in promotion focus or in prevention focus.  As shown in their own data, it is often the case that a person is high or low in both. I advise taking care in the paper to discuss these motivational/regulatory proclivities as separate, rather than opposing, influences.

The measurement of regulatory focus was also somewhat problematic, as the scales seemed to demonstrate questionable face validity. The example items included in the methods section (i.e., about getting what they want out of life or getting on their parents’ nerves) seemed largely unrelated to the theoretical constructs (using approaching or avoidant strategies). Additional information in the introduction and/or discussion regarding the measurement of these variables may be valuable.

My concerns regarding the measurement are amplified by the similarity between items in the Personal Sense of Power and Promotion Focus scales.  The example item for the former concerned being able to get what they want, while an example item for the latter concerned not being able to get their way. Ideally, a measurement model (e.g., Confirmatory Factor Analyses) would be conducted to demonstrate the relative independence and coherence of the scales prior to the tests of the theoretical proposal.

The authors correctly acknowledge the reliance on a single source as a limitation.  They suggest that the method of data collection diminished concerns regarding social desirability, and I can accept this argument.  I do not, however, feel that these steps addressed the broader concerns regarding using only self-report (e.g., the influence of other personality variables, response styles, etc.). Rather than limit their discussion of this limitation to a single sentence, a more thorough treatment of this issue, including interpreting their findings through the lens of common source variance, would improve the manuscript.

Author Response

(The authors gave the same response as above.)

Round 2

Reviewer 2 Report

The authors did a good job in addressing my previous comments.